# The World Health Organization (WHO) Integrated Care for Older People (ICOPE) Framework: A Narrative Review on Its Adoption Worldwide and Lessons Learnt

**DOI:** 10.3390/ijerph20010154

**Published:** 2022-12-22

**Authors:** Grace Sum, Lay Khoon Lau, Khalid Abdul Jabbar, Penny Lun, Pradeep Paul George, Yasmin Lynda Munro, Yew Yoong Ding

**Affiliations:** 1Geriatric Education and Research Institute, Singapore 768024, Singapore; 2National Healthcare Group, Health Services and Outcomes Research, Singapore 138543, Singapore; 3Lee Kong Chian School of Medicine, Nanyang Technological University, Singapore 308232, Singapore; 4Department of Geriatric Medicine & Institute of Geriatrics and Active Ageing, Tan Tock Seng Hospital, Singapore 308433, Singapore

**Keywords:** aged, intrinsic capacity, integrated care, person-centred care, care coordination, healthcare technology, evidence synthesis

## Abstract

The World Health Organization (WHO) recently published guidelines on the implementation of a new Integrated Care for Older People (ICOPE) framework in 2017–2019. It is an integrated care framework for the screening, assessment, and management of intrinsic capacity (IC) declines. We aimed to examine where the early adopters of ICOPE are across the world, how these study teams and sites plan to apply the framework or have applied it, and the lessons learnt for future adopters. We systematically searched electronic medical and social sciences databases and grey literature published between 31 October 2017 and 31 March 2022. Records were systematically selected using precise inclusion criteria. There were 18 ICOPE study teams and sites across the 29 selected records. Of the 18 study teams and sites, seven were in the development stage, seven conducted feasibility studies, and four have commenced implementation of interventions that applied the ICOPE framework. Future ICOPE adopters may need to make certain decisions. These include whether to adopt ICOPE in the community setting or other settings, whether to adopt only Step 1 on IC screening or additional ICOPE Steps, whether the ICOPE IC screening tool requires modifications, and whether to use digital health technology. We propose the key factors needed to make these decisions and future research needed.

## 1. Introduction

The World Health Organization (WHO) has been leading international action plans under the United Nations 2021–2030 Decade of Healthy Ageing [1]. It promotes a holistic perception of healthy ageing, which is the process of developing and maintaining functional ability to enable well-being in old age [2]. Functional ability is closely related to intrinsic capacity. The term intrinsic capacity (IC), which is the “composite of all physical and mental capacities of an individual”, was recently introduced in the WHO 2015 World Report on Ageing and Health [2]. Functional ability consists of IC of individuals, environmental characteristics, and interactions among them [3,4]. Traditional healthcare systems designed for episodic and curative care are not suitable for older persons with IC and functional ability losses or risks of losses [5]. To address this fragmentation, the WHO and recent literature propose that health and social care services should transition from siloed management of individual health conditions, towards integrated care frameworks [6,7].

The WHO Framework on Integrated, People-Centred Health Services provided a roadmap for integrating care across the life course by optimising the way services are designed, funded, managed and delivered [8]. However, a more targeted approach was needed for older adults and the WHO introduced the Integrated Care for Older People (ICOPE) framework in the end of 2017 [9,10]. Briefly, ICOPE consists of five steps (Figure 1) [11]. Step 1 in the ICOPE framework involves screening persons based on IC domains, including locomotor capacity, vitality, visual capacity, hearing capacity, cognitive capacity, and psychological capacity [11]. Individuals with IC losses proceed to Step 2 on person-centred assessment in primary care [11]. This encompasses understanding older adults’ preferences and priorities for health management, in-depth assessment of conditions linked to IC losses, assessing chronic conditions, and identifying social care and support needs [11]. Steps 3 to 5 are on developing care goals and individualized plans, linkage to specialized care and monitoring of care plans, and engaging caregivers and communities, respectively [11].

The ICOPE framework was only recently introduced. The status of ICOPE has been addressed in the publication of WHO reports on ICOPE in 2017–2022, holding WHO Clinical Consortium meetings to discuss the implementation of ICOPE, and the application of the framework by a relatively small number of early adopters globally. To elaborate, the WHO introduced the concept of ICOPE through the publication of ICOPE guidelines for community interventions in 2017 [9,10]. The guidelines provided evidence-based recommendations to healthcare providers on the approaches to detect and manage declines in physical and mental capacities at the community level. The report emphasised the need to implement the recommended guidelines using an older person-centred and integrated approach. The WHO recommendations aimed to set certain standards for national guidelines for countries [9,10]. Subsequently, the WHO published reports on implementing ICOPE in primary care, guidance for systems and services, and scaling-up mobile health technology with the mAgeing (mobile health for ageing) programme in 2018 and 2019 [11,12,13]. The reports aimed to provide further recommendations on how health and social care workers could put ICOPE recommendations into practice with or without health technology. In 2022, the WHO reported on findings from surveys from their evaluation of the readiness of countries to implement ICOPE at the clinical, service and system levels [14]. The report also included four pilot study sites that were in early phases of ICOPE implementation [14]. A recently published global Delphi consensus study examined the actions required at system and service levels for high and low-and-middle income countries (LMICs) to adopt ICOPE [15]. This study has informed the ICOPE implementation framework in the WHO report published in 2019 [12]. In addition to publishing WHO reports on ICOPE, WHO Clinical Consortium meetings were held in 2019 and 2020. These meetings reviewed the strategies to effectively move forward ICOPE implementation and identify emerging research agendas [16,17].

Given this status of ICOPE being a relatively new concept and framework, we believe that it is an opportune time for this narrative review to synthesise literature on the application of ICOPE by early adopters across the world. Our specific aims were to (1) synthesise evidence on where the early adopters of ICOPE are across the world, (2) narratively review how these study teams and sites plan to apply the framework or have applied it, and (3) propose recommendations on lessons learnt for future adopters and future research on ICOPE. In this study, all five ICOPE steps do not need to be applied for study teams to be considered to have adopted ICOPE. Additionally, modifications to the ICOPE IC screening tool were also acceptable. However, we acknowledge that the ICOPE framework consists of all steps, and screening without intervention may be more harmful than beneficial for older persons.

While existing WHO reports on ICOPE have been undoubtedly useful, this narrative review is distinct from the reports. First, the WHO reports focus on concepts on declines in IC and functional ability, promotion of the need for older person-centred care via integrated care approaches, dissemination of guidance on applying the framework at different levels of the healthcare system, and documentation of WHO meetings on ICOPE. The contribution of this review goes beyond ICOPE guidelines or updates by member countries at WHO meetings, by synthesising evidence from published records on early adopters of ICOPE globally. Some study teams and sites and individual publications in the literature may not have been captured in existing reports. Second, even though the WHO reports have incorporated case studies on ICOPE, these were focused on a few selected study sites that are piloting larger community programs and interventions. The narrative review presents a wider range of study teams and sites who are developing protocols for ICOPE programs, have conducted feasibility studies using secondary data analysis or primary data collection, and have commenced ICOPE implementation. Third, the intent and nature of this narrative review differ from reports. The WHO reports contribute to the overall agenda on the action plans on healthy ageing and documents issues raised at consortium meeting and offers guidance, whereas this narrative review aims to contribute to the academic literature and propose recommendations for practice, policy and research based on the synthesis of published records.

## 2. Methods

### 2.1. Systematic Search Strategy

The search strategy was developed by the study team in consultation with a senior medical librarian. Our search strategy applied the concepts on “Integrated Care for Older People (ICOPE)” and “World Health Organisation”. We did not combine subject headings with free texts because the concepts in the search strategy did not have Medical Subject Headings (MeSH) or other types of subject terms. Importantly, we did our best to capture the relevant records for the narrative review. Different variations of these terms were accounted for, including “World Health Organisation”, “World Health Organization”, “WHO”, “Integrated Care for Older People” and “ICOPE”. We systematically searched a total of 11 medical and social sciences electronic databases and grey literature, including PubMed, Embase, Cumulative Index of Nursing and Allied Health Literature (CINAHL), Global Health, Scopus, Google Scholar, ProQuest Dissertations and Theses Global, Global Index Medicus, WHO Reports of the Clinical Consortium for Healthy Ageing for Implementation Initiatives Across Member States, WHO Institutional Repository for Information Sharing (IRIS), and the International Conference on Frailty and Sarcopenia Research (ICSFR). Citations were uploaded to the Covidence online platform [18].

Appendix A shows the search strategies and search terms for each database.

### 2.2. Inclusion and Exclusion Criteria

Table 1 shows the inclusion and exclusion criteria based on the Study design, Population, Intervention, Comparator and Outcome (SPICO) framework.

### 2.3. Systematic Selection of Records

This narrative review aimed to timely inform decision-makers and potential ICOPE adopters on where and how ICOPE has been applied to date, and recommend lessons learnt. Hence, we used a rapid review screening protocol to accelerate and streamline the process of evidence synthesis [19]. Screening was conducted using the Covidence web-based tool [18]. Three reviewers were involved, being GS, LLK, and KAJ. First, to ensure that the reviewers were familiar with the inclusion and exclusion criteria, 20% of the titles and abstracts were screened by two reviewers. Disagreements were discussed with the third reviewer. Second, each of the remaining 80% of the titles and abstracts was screened by one reviewer; they were split among LLK, KAJ, and GS. Here, GS screened all excluded titles and abstracts to confirm exclusion. Third, each full text was reviewed by one reviewer; they were split almost equally between GS and KAJ for reviewing of eligibility. Here, LLK reviewed all excluded full text records to confirm exclusion. Disagreements were resolved by discussion to reach a consensus. Lastly, each included record was assigned to either GS, LLK, or KAJ to review bibliographies and footnote citations for records that met inclusion criteria [20]. A second reviewer confirmed that records from the bibliographic search met inclusion criteria.

### 2.4. Data Extraction and Quality Evaluation

Each selected record was divided among GS, LKL, or KAJ for data extraction. A second reviewer conducted cross-checking for accuracy and completeness. Data were extracted into structured tables. Data that the authors extracted included reference information (authors, title, year of publication), type of article, study design, country, setting, study period or length of follow-up, persons who assessed IC using ICOPE, stakeholders involved, name of program, study objective, population, sample size, inclusion and exclusion criteria of participants, and the description of the adoption of ICOPE.

Quality evaluation was not conducted. The purpose of a quality evaluation is to assess research methodology for potential biases in results. This review aimed to narratively describe the application of the ICOPE framework by early adopters and did not investigate outcomes. The omission of a quality assessment had minimal impact.

### 2.5. Data Synthesis

As mentioned, we aimed to examine how study teams and sites plan to apply or have applied the ICOPE framework. We presented the findings according to the study teams and sites across the selected records. The study teams and sites were categorised by whether they were in the: (i) development phase, (ii) feasibility phase, or (iii) implementation phase. The evaluation phase was not used in this review, as we did not aim to review the impact and effectiveness of ICOPE due to ICOPE being relatively new. If a study site that implemented ICOPE had preliminary evaluation findings, it was categorised under category (iii). This categorisation was based on the Medical Research Council (MRC) guide for developing and evaluating complex interventions [21], such as an ICOPE intervention that screens, assesses, and manages IC declines in older persons [21]. Interventions could begin at any phase, which is dependent on study objectives, contexts, and resources [21].

## 3. Results

### 3.1. Selection of Records

Figure 2 shows the PRISMA flowchart. There were 392 records identified from databases and grey literature. Screening of titles and abstracts was conducted for 317 records. Forty-eight records were assessed for eligibility and 26 met inclusion criteria. An additional three records were included from screening bibliographies. This review had 29 selected records.

Of the 29 records, study types included primary research (*n* = 14), methodology and protocols (*n* = 6), WHO reports (*n* = 3), abstracts (for full texts articles that only had English-translated abstracts, the abstracts could be included if they met the inclusion criteria) (*n* = 3), conference proceedings (*n* = 2), and a narrative article that describes the ICOPE intervention (*n* = 1). The majority were published in 2021 (*n* = 20), with the remaining records published in 2020 (*n* = 5), 2022 (*n* = 3), and 2018 (*n* = 1).

Across the 29 selected records, 18 unique study teams and sites were identified, and were in the development phase (*n* = 4), feasibility phase (*n* = 7) or implementation phase (*n* = 7). Of the 18 study teams/sites, majority described the ICOPE framework for community-dwelling persons in the general population (*n* = 13), with a smaller proportion on the primary care setting (*n* = 3) and hospital setting (*n* = 2). Majority described the use of digital health technology, such as the ICOPE app or ICOPE Monitor app (*n* = 5). Table 2 summarises the findings and Appendix A contains additional data extraction.

#### 3.1.1. Identification and Development Phase (4 of 18 Study Teams)

This phase referred to the initial conceptualisation and development of protocols of the complex intervention. Selected records were methodology and protocol studies.

#### 3.1.2. Plans to Apply ICOPE Step 1 Only

Three of the four study teams were developing Step 1 only.

The study team in Leige, Belgium presented a protocol of a prospective cohort study by Sanchez-Rodriguez (2021) that aimed to evaluate five IC domains via self-assessment using the ICOPE app and ICOPE MONITOR app at baseline, and the incidence of frailty at one-year [22]. The study will target community-dwelling persons aged 65 years and above who lived at home and were able to use mobile devices [22].

The study team in Voronezh, Russia, conceptualized the use of the ICOPE app for primary care physicians to screen patients aged 65 years and older for functional ability [23]. The study team in Costa Roca, Mexico conceptualized the use of the Longevity and Health Ageing Study Costa Rica (CRELES) dataset to conduct secondary analysis of a nationally representative sample of older adults in the community of Costa Rica [16]. Specifically, the study team intends to adopt ICOPE IC domains, but use CRELES survey questions as IC screening questions [16].

#### 3.1.3. Plans to Apply ICOPE Steps 1 to 5

One of the four study teams developed a protocol on adopting ICOPE Steps 1 to 5. It planned to target community-dwelling older persons with losses in mobility, nutritional and/or psychological domains of IC, and did not have cognitive decline or sensory impairment [24]. Biancafort Alias et al. (2021) described the conceptual framework for the Aptitude Multi-domain Group-based Intervention to Improve and/or Maintain Intrinsic Capacity in Older People (AMICOPE) longitudinal study across 11 territories in Occitanie, Andorra, Navarra and Catalonia [24]. The ICOPE programme will be supplemented or combined with components from existing interventions, such as the Vivifrail Programme for functional capacity and Feeling Well Programme for mental health, to create the AMICOPE intervention [24]. The ICOPE programme would be adopted in community facilities like senior leisure centres, civic centres, and primary care centres [24]. A multidisciplinary team would involve nurses, physiotherapists, occupational therapists, nutritionists, psychologists, and physical activity trainers [24]. Potential stakeholders included public administrators and civic organisations [24]. The next steps may be a feasibility study, followed by a RCT to evaluate the effectiveness of AMICOPE [24].

#### 3.1.4. Feasibility Phase (7 of 18 Study Teams)

The feasibility phase examined the practicality of the intervention and refined it before a full-scale implementation.

#### 3.1.5. Feasibility of Applying Only IC Domains Using Secondary Data Analysis

Three of the seven study teams conducted secondary data analysis on existing datasets to examine the prevalence of IC declines based on ICOPE IC domains. This was conducted to test the feasibility of using the IC screening domains recommended by ICOPE.

A study team in France conducted a secondary data analysis of Multi-domain Alzheimer Prevention Trial (MAPT) that was conducted from 2008 to 2013, with a 3-year follow up of each participant [25,26,27,28]. The MAPT examined the impact of a complex intervention with and without omega-3 supplements, on the prevention of cognitive decline for community-dwelling persons aged 70 years and above in memory clinics [25,26,27,28]. The MAPT was not designed to evaluate Step 1. Hence, a retrospective approach was used to define variables of interest [46]. The adapted screening tool was called the MAPT Step 1 [25,26,27,28]. Only older adults from the MAPT intervention arm had data on all IC domains [25,26,27,28]. The authors applied ICOPE screening questions to assess the domains on cognition, locomotion, vitality, and nutrition on the participants in the MAPT dataset [25,26,27,28]. However, the team was unable to use the exact ICOPE screening questions under the hearing, vision, and psychological domains. Instead, the authors used retrospective data on self-reported visual acuity), the validated hearing handicap inventory for the elderly (HHSE-S), and items 2 and 7 of the Geriatric Depression Scale (GDS-15) [25,26,27,28].

A study team conducted secondary analysis of the Mexico Health and Ageing Study (MHAS) Wave 2015 on individuals aged 50 years and older in the Mexican population [29]. Another study team analysed the dataset from the Dementia Research Group (DRG) cohort study (2003–07, 2008–10) in LMICs across Latin America, China and India on community-dwelling persons aged 65 years and above [30]. Variables from these two surveys, which were not designed to assess IC, were selected to construct the teams’ own screening tools based on ICOPE IC domains [29,30]. The latter study team did not assess a hearing domain [30].

#### 3.1.6. Feasibility of Implementing Step 1–3, with Modifications to the ICOPE Screening Tool

One of the seven study teams examined the feasibility of implementing an adapted version of Step 1.

Won et al. (2021) was a six-month feasibility study of the Integrated Care of Older Patients with Frailty in Primary Care (ICOOP_FRAIL) at four primary care clinics [31]. The ICOOP_FRAIL was the first integrated care programme for frailty and functional decline in primary care in South Korea [31]. Nurses and doctors applied the Korean Frailty Index for Primary Care (KPI_PC) on mobile devices to screen and monitor IC at three timepoints [31]. The ICOOP_FRAIL applied the ICOPE IC screening domains into the 53-item KPI_PC, and modified the IC screening questions for practicality and to suit the local context [31]. Instead of the hearing test, the team used self-reported problems with hearing by patients; instead of self-reported eye diseases or receiving treatment for diabetes or hypertension, patients only self-reported poor eyesight; instead of self-reported falls and abnormal gait and/or balance, patients were assessed for number of falls, use of walking aids, and issues with balance as evaluated by physicians [31]. Steps 2–3 involved patient-centred assessments and interventions by health coaches [31].

#### 3.1.7. Feasibility of Implementing Step 1, without Modifications to the ICOPE Screening Tool

Three of the seven study teams investigated the feasibility of adopting Step 1, and the records did not mention modifying the ICOPE screening tool.

Ma and colleagues published cross-sectional feasibility studies in Beijing, China [32,33]. The authors examined the clinical utility of Step 1 on healthy older adults without acute illnesses in the department of geriatrics of a University hospital [32,33].

Another cross-sectional feasibility study on Step 1 was conducted in the villages of Jodphur, Rajasthan, India on community-dwelling older persons [34]. The ICOPE programme would be one of the first interventions to provide patient-centred care plans [17]. Self-screening with mobile technology was not feasible due to low literacy rates [14]. Interviewers, including public health students, conducted screening at the homes of individuals aged 60 years and above [34]. There may be plans to evaluate Steps 2 to 5 after the lifting of COVID-19 restrictions [14,16,17]. It may involve modifying Step 1 to suit colloquially used syntax and vocabulary, and using the Hindi Mental State Examination [17]. Some healthcare workers at Rajasthan had limited knowledge and interest in the care of older persons. The study team also pointed out that the lack of funding was a barrier to scale the implementation of ICOPE.

A feasibility study was conducted at a University hospital in Nice, France, on older patients with dementia and their caregivers [35]. Nurses used the ICOPE MONITOR app, and referred patients with IC declines to physicians, pharmacists and psychologists for further assessment and intervention [35].

#### 3.1.8. Implementation (7 of 18 Study Sites)

Seven of 18 study sites were in the implementation phase, which referred to the application of the intervention in real world settings.

#### 3.1.9. Implementation of Step 1 Only

Three of the seven study sites only applied Step 1.

A cross-sectional study in Taiwan was conducted on community-dwelling adults aged 75 years and older, or aged at least 65 years with hypertension, diabetes or hyperlipidaemia [36]. It aimed to examine associations between IC with chronic conditions [36]. Liu et al. (2021) reported a two-year cohort study on persons aged 75 years and above at a retirement community in Beijing, China [37]. The ICOPE IC screening domains in Step 1 were applied to data within the Comprehensive Geriatric Assessment (CGA) Electronic Data Capture System by geriatricians from the Peking Union Medical College Hospital [37]. Yu et al. (2022) aimed to examine the prevalence and distribution of IC in older adults at 80 community elder centres across 18 districts in Hong Kong [38]. Older persons did self-screening with guidance from staff [38].

#### 3.1.10. Implementation of Steps 1–3 Only

Two study sites implemented only Steps 1–3.

A study team in Ohio, US, implemented ICOPE across 45 primary care group practices. Step 1 primarily aimed to identify older adults with or were at risk of nutritional conditions [39]. Steps 2–3 involved further assessment by general practitioners to recommend supplements and provide educational resources on nutrition [39].

At a study site at Canillo, Andorra, geriatricians and a geriatric nurse adopted Step 1 for community-dwelling persons aged at least 65 years [14]. Recruitment was done via a public health media campaign [14]. Screening was conducted at a social club for older persons run by the city council to identify IC declines [14]. Step 2 involved IC assessment at a community health facility regardless of results from Step 1. Step 3 involved development of individualized care plans by geriatricians and the primary care doctors followed up [14]. A barrier highlighted by the study team was human resource constraints. The study team was small and had to work within time constraints.

#### 3.1.11. Implementation of Steps 1–5

Two of the seven study sites implemented all five steps [14,16,17,40,41,42,43,44,45,46].

The INSPIRE research initiative consisted of two human cohorts with a 10-year study period (2019–2028), including the INSPIRE Human Research Translational cohort (INSPIRE-T cohort) and INSPIRE ICOPE-Care cohort [16,17,41,46]. The former examined ageing biomarkers among persons aged 20 years and above in the community and those institutionalized [17]. The latter involved the pragmatic implementation of ICOPE in the health system, mainly in Toulouse City [17,42,43]. Older adults in the community and those who utilised primary care services were recruited. The INSPIRE ICOPE-Care programme involved Step 1 screening every 4–6 months via self-assessment and/or healthcare workers, including community nurses, pharmacists, and general practitioners [40,41,46]. The programme aimed to screen and monitor 200,000 older persons in Occitanie in the first three to five years [46], and recruited 10,900 participants with mean age 76 years from January 2020 to November 2021 [14]. The use of the ICOPE app, ICOPE MONITOR app, a conversational chatbot, and the FRAILTY-ICOPE telemedicine database allowed remote and large-scale IC screening and monitoring [14,16,17,40,41,42,43,44,45,46]. Digital tools had algorithms to alert healthcare workers when participants had IC declines [46].

Steps 2–3 involved CGA and biological sampling by primary care workers, including physicians, nurses trained in geriatrics and physiotherapists, and development of care plans [17,44]. Nurses and pharmacists had access to free webinar trainings [45]. Step 4 entailed using mobile health technology and the FRAILTY-ICOPE telemedicine database to remotely monitor a large number of participants, provide linkage to medical resources and follow-up on care plans [14,16,17,40,41,42,43,44,45,46]. Teleconsultation by healthcare professionals with training in geriatrics were conducted for some participants [17]. Step 5 involved community engagement, whereby the study team was in the midst of establishing an ecosystem among councils, town halls and organisations to promote healthy ageing [14,46]. A long-term plan was for ICOPE-Care to be scaled-up to other parts of France [46]. Primary care doctors at Occitanie had limited time and insufficient reimbursement to support ICOPE assessments and interventions. The study team highlighted the lack of awareness among primary care physicians on the opportunities to slow the decline of IC of older persons. Only IC screening was covered by health insurance, hence, a barrier was a lack of financial incentive within the healthcare system to integrate the latter steps of the care pathway.

The ICOPE programme at Chaoyang, Beijing, China aimed to promote healthy ageing in community-dwelling older adults, with the slogan translating to “ICOPE will bring older people happiness” [16]. Step 1 involved screening for deficits in IC domains at homes and health centres [14,47]. Steps 2–3 were assessments by integrated care managers who were trained health and social care workers, and development of care plans [14]. Steps 4–5 involved integrated care managers having tele-consults to review medication adherence, rehabilitative exercises, and linkage to additional health and social services [14]. It commenced with a one-year longitudinal study in 2021, among approximately 7000 participants aged 65 years and older [14,17,47]. More than 22,000 physicians, nurses, rehabilitation therapists, and social workers have indicated interest in the study [14]. Stakeholders included over 200 organisations, such as community health centres, the Ministry of Civil Affairs, Beijing Health Commission, and the WHO China Office [14]. At Chaoyang, there was a low ratio of integrated care managers to participants. The low number of primary care physicians may not have the manpower and skills to manage the complex cases, and engaging specialists who were not geriatricians was challenging.

## 4. Discussion

### 4.1. Lessons Learnt and Implications for Practice and Policy

Early adopters of care models play an important role in informing their development and application for stakeholders [48,49]. Our findings have implications for health policymakers and practitioners who are considering adopting ICOPE. To the best of our knowledge, this is the first review in the literature that synthesises evidence on the adoption of ICOPE globally since ICOPE guidelines and reports were circulated in 2017–2022. There is a limited ability to compare findings with existing literature. This review still contributes to the literature by synthesising evidence on ICOPE adoption across the world, discussing the framework, and providing recommendations for practice, policy, and research. We identified five key decisions that adopters may need to make:1Should the ICOPE programme target community-dwelling older adults in the general population or only older adults who attend primary care?

The objectives of ICOPE are to prevent IC declines and maintain IC in older adults in the community. Early adopters need to decide which group of older adults in the community to undergo IC screening, what is the first step prior to comprehensive assessment, developing individualized care plans and linkage to specialized care. Specifically, early adopters may need to target community-dwelling older adults in the general population or only community-dwelling older persons who attend primary care. Current evidence on adoption is largely for community-dwelling older adults who did not need to have interactions with primary care. There may be operational and logistical advantages to recruiting community-dwelling older persons in the general population via media campaigning and social media [14]. In contrast, there is a smaller pool of older persons for recruitment in a primary care setting, and effort is needed by healthcare workers to facilitate recruitment [22,31].

Another factor to consider is intervention objectives. For instance, studies that targeted community-dwelling older adults in the general population aimed to estimate the prevalence of IC declines in the community, to have early detection of IC decline, or to study the development of IC deficits over time in a large population [41,47]. Studies that focused only on recruitment and IC screening in the primary care setting aimed to improve IC detection and have targeted intervention and management measures for primary care patients [22,39].

2Whether to adopt only Step 1 on IC screening or include other steps? What are the considerations for adopting ICOPE partially?

From the synthesis of the literature, it appears that ICOPE has been adopted partially by an appreciable proportion of study teams. Specifically, some study teams only applied Step 1 and some only applied a subset of the six IC domains in Step 1. Hence, potential early adopters need to assess the ICOPE framework and be made aware of some caveats.

First, potential adopters may need to consider where the current gaps are in their health systems. Adopters from locations with integrated care models and health system infrastructures that are similar to Steps 2–5 may consider only applying Step 1. Alternatively, they could consider integrating this IC assessment within their existing CGA. In contrast, study sites with health system gaps in both IC screening and integrated care could adopt at least Steps 1–3. For instance, study sites in South Korea and India that aimed to adopt new integrated care models adopted downstream ICOPE steps [14,31,34].

In addition, for adopters who already have care processes (e.g., CGA) in place and intend to employ them in lieu of Steps 2–5, the ICOPE framework is potentially still useful in informing what care elements are needed or how existing ones can be improved in their current care systems. Hence, the entire ICOPE framework could still value-add towards health systems that have adopted CGA or equivalent care processes in primary care and the community.

Second, assessment of IC domains may already be applied in practice, prior to these domains being conceptualized and known as ICOPE IC screening. This was shown by the study teams who were able to conduct retrospective analysis on existing clinical data to examine the prevalence of IC declines based on IC domains [25,26,27,28,29,30]. Screening programmes in the community or CGA in clinical practice may have incorporated parts of the six IC domains. Hence, adopters may appear to apply ICOPE partially, by only adopting IC domains that were previously not part of their existing assessment program.

Third, the measurement tools recommended by WHO for IC screening may not be available or applicable in the contexts of adopters. This may limit the ability of adopters to apply ICOPE IC screening in the exact form that was recommended in the WHO reports and ICOPE app. The IC domains are broadly defined and there are currently no alternative measurement tools that have been recommended. This may result in study teams being deterred to adopt ICOPE or to operationalise it differently. While it is beneficial to have flexibility on how to apply IC screening and the use of a range of measurement tools, there will be challenges in the comparison of IC prevalence across study sites and validation of IC measurements. Lastly, even if study teams identify the need to adopt Steps 2–5 and go beyond Step 1, a barrier is resource constraints. These resources include sufficient manpower, training of doctors, community nurses and non-healthcare staff, and established healthcare eco-systems to facilitate coordinated referral pathways from primary care to specialist care.

Furthermore, integrated care should encompass screening with subsequent comprehensive assessment and intervention. If the care pathway does not include further assessment and interventions after IC deficits are detected at screening, it may be more detrimental than beneficial for older persons. Additionally, the assessment of feasibility should ideally be on the entire care pathway, including IC screening, CGA, public health interventions, monitoring, and follow up. While ICOPE early adopters could start by examining the feasibility of adopting Step 1, it is ideal to examine the feasibility of measures downstream from screening.

3Whether to modify the IC screening tool in Step 1?

Whether the ICOPE IC screening questions need to be modified depends on the target population and local contexts. For older adults who could not administer the auditory and visual tests themselves or by healthcare workers, the sensory domain may have to be modified to self-reporting of hearing and visual problems [25,27]. Some study sites modified the IC screening questions to contextualize them to target populations, such as using colloquial syntax and vocabulary [14,34], having physicians assess balance and use of walking aids instead of patients self-reporting mobility deficits [31], or using locally validated tools on depressive symptoms [14,34].

In addition, if a study team intends to apply the ICOPE Step 1 screening domains in existing data collected from previous surveys or clinical trials [29,30], then there is a high likelihood that the IC screening questions will need to be based on existing datasets. The exact screening questions of the ICOPE IC screening tool may not be used.

4Whether to use digital tools

This depends on the aims of the study teams, such as whether large-scale data tracking is required. Mobile health technology facilitated ICOPE adoption for sites that aimed to detect and monitor IC for large sample sizes at multiple time points, like the INSPIRE ICOPE Care program. The programme aimed to have long-term and regular follow-up of 200,000 older adults who used mobile apps and a conversational chatbot that guided their self-screening [43,46]. Telemedicine databases facilitated the monitoring of IC trends, tracking those who needed follow-up from IC declines, and large-scale analysis [43,46]. A benefit of using the ICOPE app is to encourage users to track their own IC declines.

Another consideration is that the ICOPE app will not be applicable if study teams want to modify some of the IC screening questions. 

An important factor is the technological ability of the target population. Older adults in Rajasthan, India, were not able to self-administer the screening tool due to illiteracy [14,34]. Other study sites’ inclusion criteria were limited to only those who could use mobile devices [22]. Finally, technological infrastructure will be needed for tracking data of large numbers of persons over time, data protection, and technical support [50,51,52].

5At which phase of the intervention should ICOPE be adopted?

This refers to deciding whether to adopt ICOPE in the (a) feasibility phase (only IC screening; secondary data analysis), (b) feasibility phase (with primary data collection), (c) implementation phase.

Factors to consider include resources and study aims. Most sites at the feasibility phase aimed to examine IC prevalence by applying ICOPE IC domains within existing data collected on their target populations [25,26,27,29]. These were resource-efficient methods to test the feasibility of applying the ICOPE IC domains in existing datasets before implementation and primary data collection. Hence, potential adopters with resource and time constraints could apply the same methods on local datasets to examine its feasibility of assessing the prevalence of IC declines.

Comparatively, early adopters that went into the implementation phase appeared to have the resources to conduct large-scale programmes. Implementation of the INSPIRE ICOPE-Care programme was part of a major long-term research initiative involving laboratory animals and people to understand biomarkers, mechanisms for ageing, and improving clinical care [43]. Their study objective was also to implement large scale interventions to target IC, and not only to understand prevalence of IC decline.

### 4.2. Implications for Future Research

First, we could compare programmes that adopted Step 1 with little to no modifications, with those that adopted ICOPE IC domains and modified the screening questions. These comparisons could include prevalence of IC declines detected, and the feasibility and effectiveness of implementation. We are of the opinion that harmonising the IC screening tool will be challenging, due to cultural and contextual differences across study sites worldwide. There is no standardised screening tool or process for the assessment of other geriatric conditions, such as frailty [53,54]. Similarly, various measurement tools have been used to assess IC, particularly in the vitality and psychological domains [55].

Second, with most study teams focused on Step 1, we recommend more studies on Steps 2–5. We need to better understand downstream components of ICOPE on integrated care and engagement of communities and caregivers that follow the detection of IC declines. Additionally, studies could test our hypothesis mentioned in the preceding section that early adopters with existing health system infrastructures for downstream measures of integrated care would benefit from focusing on Step 1 or integrating IC domains within existing CGA, while those with health system gaps in both IC screening and integrated care would benefit from adopting beyond Step 1. Future studies could examine the resources and infrastructure needed to implement ICOPE beyond Step 1. This review appears to show that larger urban settings may facilitate integrated care after IC detection [42,47].

Third, only a few studies have reported preliminary findings and these findings were primarily related to the proportion of participants with IC declines detected during screening, and socio-demographic and health profiles of those with IC declines. Future research is needed to fill this dearth in the literature, by going beyond the examination of prevalence of IC declines to explore the impact and effectiveness of ICOPE. As more study sites progress towards the advanced stages of implementation, there could be systematic data collection on clinical and patient-reported health outcomes and cost-effectiveness. Qualitative findings from patients and implementers on facilitators and barriers could tease out mechanisms of change for this complex intervention.

Fourth, some study sites in the implementation phase mentioned that a barrier to ICOPE implementation was related to the limited size, skills, and incentives of the workforce. Future implementation studies could examine how to better tackle workforce challenges, such as manpower and time constraints, lack of knowledge and skills to conduct the downstream assessments and interventions after IC screening and limited financial incentives for healthcare and non-healthcare workers. 

Lastly, further research could investigate which settings, subgroups of older adults, and type of health and social care delivery systems may enhance the adoption and effectiveness of ICOPE.

### 4.3. Limitations

Three records only had data extracted from English-translated abstracts. Due to the lack of access to translation resources, information from full texts were not extracted which could have provided further information. However, this practice has become acceptable, and evidence suggested it may not significantly bias review findings [56]. 

Despite our best efforts to search databases and several grey literature sources, and conduct bibliographic searches of included records, we acknowledge that relevant records may be omitted. However, based on three WHO reports on key ICOPE study sites, all sites were covered by our selected records [14,16,17]. The authors had to balance between the time and manpower resources spent on conducting the search and publishing a timely narrative review for potential early adopters. Current study sites are in relatively preliminary stages of the ICOPE implementation. It would take more time for study sites to publish more findings on the effectiveness and outcomes of the ICOPE program. When this happens in the future, we acknowledge that a formal systematic review on the effectiveness of the implementation of ICOPE programs will benefit from a more comprehensive search.

There may be a lack of clarity on the decision-making process by study sites on their adoption of ICOPE Step 1 and reasons for not adopting more downstream steps, which could elicit further insights. Lastly, a review of outcomes was beyond the scope of this review and limited by the dearth in the literature on the impact of ICOPE.

## 5. Conclusions

This narrative review described where the early adopters of ICOPE are across the world, how these study teams and sites planned to apply the framework or have applied it, and the lessons learnt for future adopters. The study teams and sites are mostly in the development and feasibility phases, and the early implementation of ICOPE. Future adopters may need to make certain decisions. These include whether to adopt ICOPE in the community setting for the general population or other settings, whether to adopt only Step 1 on IC screening, whether the IC screening tool requires modifications, and whether to use digital health technology. This review suggested the key factors needed to make these decisions and the future research needed.

## Figures and Tables

**Figure 1 ijerph-20-00154-f001:**
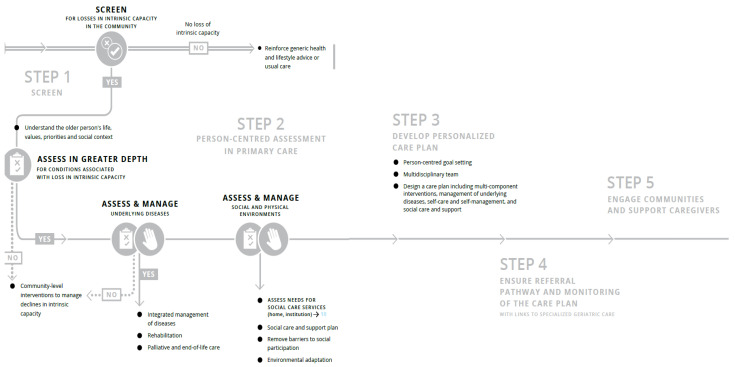
ICOPE Steps 1 to 5. Derived from [11].

**Figure 2 ijerph-20-00154-f002:**
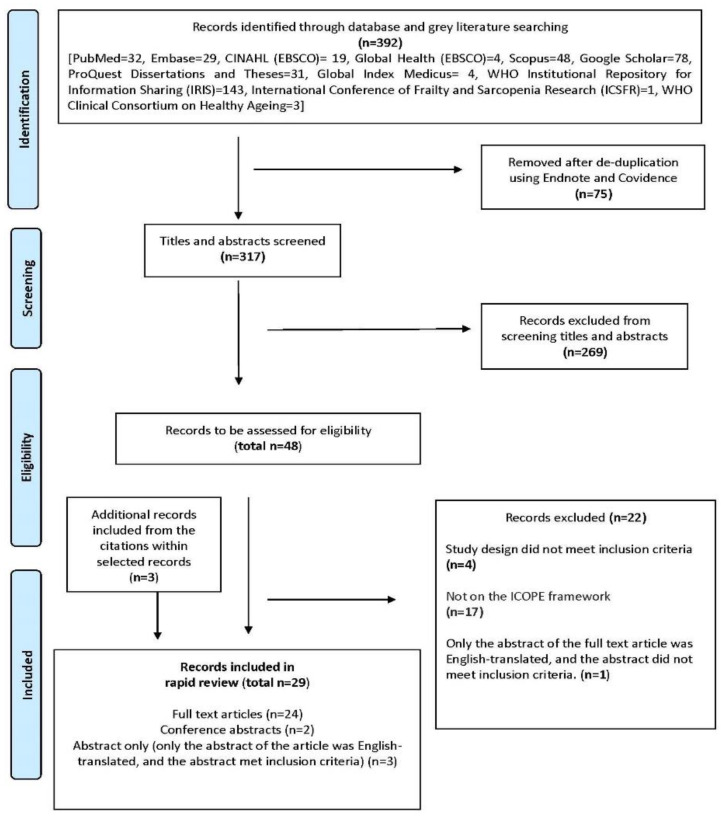
PRISMA flowchart.

**Table 1 ijerph-20-00154-t001:** Inclusion and exclusion criteria for record selection.

Inclusion		Exclusion
**Study design**		
1.	Primary and secondary peer-reviewed articles Protocols and methodology articles. Grey literature (conference proceedings, pre-prints, dissertations, government reports, policy reports)	Editorials or letters to the editor Commentaries Concordance testing and prediction models
2.	Published from 31 October 2017 to 31 March 2022.The WHO introduced the WHO ICOPE framework by publishing a WHO report on the ICOPE guidelines for community interventions in October 2017. Hence, we searched for records published from 31 October 2017. The search was conducted in early April 2022, hence, we searched for published records up till 31 March 2022.	Published before 31 October 2017 or after 31 March 2022.
3.	In English or with English translations. We contacted the authors of non-English articles and engaged an information specialist to confirm that the records had no English translations. Abstracts from conference proceedings in English or with English translations could be included if it met the inclusion criteria of adoption of ICOPE. For full text articles that only had English-translated abstracts, the abstracts could be included if they met the inclusion criteria of adoption of ICOPE.	Not in English and had no English translations.
**Population**		
4.	Older adults	
**Intervention**		
5.	We aimed to find literature that answers the research questions: Where are the ICOPE study teams and study sites around the world? How do these study teams and sites plan to apply or have applied it? Hence, the inclusion criteria were: The record mentions that the ICOPE framework was applied. All five ICOPE steps do not need to be applied. If only ICOPE Step 1 on IC screening is applied, the record can be included. Modifications to the ICOPE screening tool for IC can be accepted. Methodology and protocol articles that describe how the ICOPE framework was planned to be applied can be included.	The record is not on the ICOPE framework.
**Comparator**		
6.	Records with and without controls can be included	
**Outcome**		
7.	There were no inclusion criteria on outcomes. This was supported by: Implementation of ICOPE being in the preliminary stages since the publication of WHO ICOPE guidelines in the end of 2017–2019. The review focused on synthesising literature on the application of ICOPE to date, and not on its effectiveness. Impact and effectiveness of ICOPE will be more meaningful to investigate later, after more study teams and sites have implemented it and for a longer time.	

**Table 2 ijerph-20-00154-t002:** Summary of the 18 ICOPE study teams and sites.

Phase of Intervention		Citation	Study Participants	Who Does IC Screening	Brief Description of Application of the ICOPE Framework
**Development** **phase** **(4 study teams)**	**Study teams that plan to adopt ICOPE Step 1 only** **(Methodology and protocol articles)** **(3 study teams)**	**Study team 1**			
		**Methodology of a 1 year prospective cohort study in Leige, Belgium** **Setting: Community** **Mobile health technology: ICOPE app, ICOPE MONITOR app**			
		Sanchez-Rodriguez 2021 [22]	Community-dwelling adults aged 65 years and above who are living at home and able to use apps.	Self-assessment using two apps, the ICOPE app and ICOPE MONITOR.	The ICOPE app will assess five IC domains. Two results are possible: positive (probable decrease in intrinsic capacity) or negative (intrinsic capacity not decreased), as a total binary result of the five domains together. The app includes the possibility to record the summary of the screening, download it in PDF, or send it by mail. The ICOPE MONITOR app will be used for identification, obtaining informed consent of both healthcare workers and the participants, and monitoring five IC domains. The results obtained in the five domains will be provided as a checklist: two results are possible for each one of the five domains: Yes or No in each domain. Recommendations for each domain are provided. The app automatically schedules the date for the next IC screening every 4 months and sends the informed consent by mail to participants.
		**Study team 2**			
		**Development of a study in Voronezh, Russia** **Setting: Primary care** **Mobile health technology: ICOPE app**			
		Alekhina et al., 2021 [23]	Patients aged 65 years and above.	Primary care physicians.	Physicians screen for functional ability using a mobile app.
		**Study team 3**			
		**Plans to conduct secondary analysis of the CRELES (2005–2009) dataset** **Nationally representative sample in Costa Rica, Mexico**			
		WHO Report 2020 [11]; Mexico	CRELES dataset consists of a nationally representative sample.	Not specified.	Among the variables used in the study was the IC index, which is a summary index built from the five ICOPE domains using a scale of 0 (severe deterioration) to 10 (optimal).
	**Study team that plans to adopt ICOPE Steps 1–5**	**Study team 1**			
		**Conceptualisation of the AMICOPE Study in Spain and France (11 territories in Occitanie, Andorra, Navarra, Catalonia)** **Setting: Community**			
		Blancafort Alias et al., 2021 [24]	Older people with losses in mobility, nutritional and/or psychological domains of intrinsic capacity; without cognitive decline, visual impairment, or hearing loss; living in the community; and recruited or referred from primary care and community settings.	Healthcare workers	Conceptual framework involved multimodal exercises, advice on cognitive stimulation, dietary advice, advice on managing depressive symptoms like reducing stress, tackling loneliness, strengthening social support, and accessing community facilities, medication review, supporting self-management, goal setting, and maintaining autonomy and preventing dependence on care. Older persons will be recruited from community facilities, such as senior leisure centres, civic centres, or primary care centres. Health and social care professionals would be from different backgrounds, including nurses, physiotherapists, occupational therapists, nutritionists, psychologists, and physical activity trainers.
**Feasibility phase** **(7 study teams)**	**Feasibility of only applying ICOPE IC domains using secondary data analysis** **(3 study teams)**	**Study team 1**			
		**Secondary analysis of the MAPT dataset** **Setting: Community (memory clinics in the community in France)** (MAPT is a 3-year, multi-centre, randomised, placebo-controlled superiority trial)			
		González-Bautista et al., 2021 [25] (J Frailty Aging, Frequency of conditions)	Community-dwelling older adults aged 70 years or older without dementia at baseline.	Secondary data analysis of the MAPT dataset.	Step 1 is called MAPT Step 1. The IC domains included cognitive decline, limited mobility, malnutrition, visual impairment, hearing loss, and depressive symptoms, and they refer to the conditions associated with declines in IC. Adapted operationalisation of: Visual impairment: Self-reported visual acuity items. Hearing loss: Item number 3 of the screening version of the hearing handicap inventory for the elderly (HHSE) Depressive symptoms: Defined according to items 2 and 7 of the Geriatric Depression Scale (GDS-15), which were judged by three experts (one geriatrician, one general practitioner, and one researcher in clinical gerontology) as being the most similar items compared to those recommended by WHO. After being screened with the adapted MAPT Step 1 tool, approximately 9 of 10 older adults had one or more conditions associated with declines in IC.
		Gonzalez-Bautista et al., 2021 [26] (Ageing Clinical Experimental Research Journal)			Only the cognitive ICOPE IC domain was applied. People with impaired cognitive capacity according to the ICOPE Step 1 tool had higher dementia risk.
		Gonzalez-Bautista et al., 2021 [27] (Maturitas Journal)			Step 1 is called MAPT Step 1. The study appliedthe modified ICOPE Step 1 for 3 domains, including cognition (time and space orientation plus word recall), locomotion (perform five chair rises within 14 s), and vitality/nutrition (self-reported weight loss or appetite loss). Adapted operationalisation of: Vision: answering “yes” to any of: “Even if wearing glasses, do you have visual problems to (a) distinguish the faces of people in the same room? (b) move indoors/outdoors? (c) other activities (reading a paper, watching television)?” Hearing: answering “sometimes” or “yes” to the question “Do you have difficulty hearing when someone speaks in a whisper?” (HHSE-S); Psychological function: answering “yes” to the item 2 of Geriatric Depression Scale (GDS-15) “Have you dropped many of your activities and interests?”, or responding “no” to the item 7 of the GDS-15 “Do you feel happy most of the time?” Three experts (one geriatrician, one general practitioner, and one researcher in clinical gerontology) judged these GDS items as the closest ones to the ICOPE screening. Calculated total score by adding the number of intrinsic capacity impairments found by the MAPT Step 1 (score range 0–6, higher is worst). Decline in each additional IC domain was associated with a higher risk of incident frailty, IADL, and ADL disability.
		Pages et al., 2022 [28]			ICOPE Steps: Step 1 (modified, referred to as MAPT Step 1) Applied an adapted version of the ICOPE Step 1 to screen for 6 IC domains.
		**Study team 2**			
		**Secondary analysis of the Mexico Health and Ageing Study (MHAS) Wave 2015 dataset;** **Nationally representative population**			
		Gutierrez-Robledo et al., 2021 [29]	Older adults aged 50 years and older in the Mexican population.	Secondary data analysis of an existing dataset.	The IC domains included cognition, psychological, senses (vision and hearing), vitality, and mobility. The exact ICOPE screening questions may not be used. The questions will be from the MHAS survey. Decreased levels of IC in the Mexican older people were associated with lower levels of education, poorer self-rated health, more chronic diseases, more visits to a physician, and increased dependencies in ADLs.
		**Study team 3**			
		**Secondary analysis of the Dementia Research Group (DRG) cohort study (2003-07, 2008-10) dataset on 8 LMICs across Latin America, China, and India** **Setting: Community**			
		Prince et al., 2021 [30]	Older people aged 65 years and over living in geographically defined catchment areas in eight countries.	Interviewer who visited participants’ homes.	The IC domains were evaluated at baseline, including neuromusculoskeletal capacity, vitality, nutrition, sensory capacity (visual impairment), cognitive capacity, psychological capacity, and continence. Declines in IC was associated with dependence for care and mortality.
	**Feasibility of implementing Step 1, with modifications to the ICOPE IC screening tool** **(1 study team)**	**Study team 1**			
		**6-month feasibility study of ICOOP_FRAIL in South Korea (A RCT across 4 primary care clinics)** **Setting: Primary care** **Mobile health technology: IC screening (KPI_PC) on mobile devices**			
		Won et al., 2021 [31]	Not specified.	Nurses and doctors.	Nurses and doctors applied the Korean Frailty Index for Primary Care (KFI_PC) on mobile devices to evaluate the functional decline and frailty of older adult patients. The KFI_PC consists of 53 items in eight IC domains, including cognition, mobility, malnutrition, visual impairment, hearing loss, depressive symptoms, and geriatric syndromes including urinary incontinence and risk of falls. All items in the KFI_PC were on a mobile notepad used by nurses and doctors to evaluate the frailty index score and assessed risk factors for frailty. Primary care physicians talked to patients about functional declines and its risk factors, nutrition, exercise, medication, and disease management. Within 1 month, a health coach contacted the patient via telephone to monitor them and encouraged consuming a protein-rich balanced diet or/and exercise. The patient was linked to social welfare services managed by community administration centres or senior welfare centres for social support. However, none were available due to COVID-19. Health coaches involved in ICOOP_Frail were paramedical students. The phone call for health coaching was repeated monthly for a total of 6 months. The KFI_PC was re-evaluated at 3 and 6 months
	**Feasibility of implementing Step 1, without modifications to the ICOPE IC screening tool** **(3 study teams)**	**Study team 1**			
		**Cross-sectional study by a team in Beijing, China** **Setting: Hospital setting (Department of Geriatrics of a University Hospital)**			
		Ma et al., 2020 [32]	Healthy participants without acute illness and aged ≥50 years.	Not specified.	The authors examined the clinical utility of Step 1 on healthy older adults without acute illnesses in the department of geriatrics of a University Hospital. The ICOPE screening tool appeared useful to identify adults with poor physical and mental function in a Chinese sample.
		Ma et al., 2021 [33]			
		**Study team 2**			
		**Cross-sectional feasibility study in the villages of Jodphur, Rajasthan, India** **Setting: Community**			
		Mathur et al., 2021 [34] (preprint)	Geriatric persons aged 60 years and above.	Community-based healthcare workers and trained interviewers, including trained public health students	The IC domains included cognition, mobility, visual and hearing, and depressive symptoms. The ICOPE screening tool provided the proportions of older persons with declines in each of the IC domains.
		WHO Clinical Consortium on Healthy Ageing 2019 [16]			It will involve ICOPE screening, assessment and interventions. Follow-up period is planned to be 15 months.
		WHO Clinical Consortium on Healthy Ageing 2020 [17]			Step 1 will be by community-based health workers. Step 2 will be by mid-level health providers and doctors. Step 3 will be by the multidisciplinary team. Step 4 (Referrals) will be by hospital care navigators to the appropriate assessments and interventions, and Step 5 is on caregivers. Community health workers, case managers and navigators (trained to assist patients in seeking the right level of care in hospital settings) will help align health and social services and facilitate referral of rural older adults.
		ICOPE implementation pilot programme: findings from the ‘ready’ phase. Geneva, 2022 [14]			Step 1 was conducted at the homes and the community in Rajasthan by 15 public health students. Assessments were planned. Due to COVID-19, no further activity was possible following IC screening. The most significant declines were in hearing and mobility, with more than half of the older persons having declines in these domains. There were higher levels of decline in the mobility domain among female than male participants.
		**Study team 3**			
		**6-month feasibility study by a study team in Nice, France** **Setting: Hospital** (University Hospital Centre of Nice) **Mobile health technology:** ICOPE MONITOR app			
		Valliant-Ciszewicz et al., 2021 [35] (Conference proceeding)	Dementia patients and their caregivers (patient-carer dyads).	A nurse	Nurses carried out Step 1 at the first month and fourth month of care. Nurses carried out screening for frailty and prevention activities based on the ICOPE MONITOR app. For Step 2, the nurse referred older adults to hospital prescribers and/or attending physicians, if declines in IC were identified. Interventions by geronto-psychologists were given at home, including non-drug approaches for the patient and psychoeducation programme for the carer.
**Implementation phase** **(7 study teams/sites)**	**Implementation of Step 1 only** **(3 study teams/sites)**	**Study team/study site 1**			
		**Cross-sectional study by a study team in Chia-Yi, Taiwan** **Setting: Community**			
		Cheng et al., 2021 [36]	Community-dwelling persons aged at least 65 years with hypertension, diabetes or dyslipidemia; or persons aged at least 75 years.	Not specified	The team aimed to examine associations between IC with chronic conditions IC domains included cognitive decline, limited mobility, malnutrition, visual impairment, hearing loss, and depressive symptoms. Dyslipidemia was associated with greater declines in IC.
		**Study team/study site 2**			
		**2-year cohort study in Taikang Yanyuan, Beiing, China** **Setting: Community (Taikang Yanyuan continuing care retirement community)**			
		Liu et al., 2021 [37]	Older adults aged 75 years and more who lived in the compound senior community. They were independent, as evaluated by community doctors.	Trained geriatricians from Peking Union Medical College Hospital (PUMCH).	The IC domains included cognition, locomotion, vitality, sensory capacity (vision and hearing) and psychosocial needs. Step 1 screening was adapted by using the Comprehensive Geriatric Assessment (CGA) Electronic Data Capture System. IC decline was more strongly associated with adverse outcomes than with frailty.
		**Study team/study site 3**			
		**Longitudinal observational study of 18 districts in Hong Kong** **Setting: Community (80 community elderly centres across 18 districts in Hong Kong.)**			
		Yu et al., 2022 [38]	Community-dwelling older adults	Older adults completed IC surveys with guidance from trained staff members of the elderly centres.	Six IC domains included cognition, locomotion, vitality, sensory capacity (vision), sensory capacity (hearing), psychological capacity. The team aimed to examine the prevalence and distribution of IC in older adults at 80 community-based elder centres across 18 districts in Hong Kong. IC decline was associated with increased risks of polypharmacy, incontinence, poorer self-rated health, and IADL dependencies.
	**Implementation of Steps 1–3** **(2 study teams/sites)**	**Study team/study site 1**			
		**Longitudinal observational study in Ohio, United States** **Setting: Primary care (45 primary care group practices)**			
		Smith 2018 [39]	Older adults with nutritional conditions or at risk of nutritional conditions.	Primary care practitioners	Step 1 involved identification of nutritional conditions. During routine office visits, primary care practitioners used nutrition care pathways to identify the potential of being under- or overnourished. Practitioners recommended oral nutrition supplements appropriate for the patient’s condition and provided basic nutrition education through a variety of resources. Follow-up care was given during subsequent patient office visits.
		**Study team/study site 2**			
		**Observation study from July–Sept 2020 at Canillo, Andorra** **Setting: Community (Small urban site in a small town in a mountainous area)**			
		ICOPE implementation pilot programme: findings from the ‘ready’ phase. Geneva [14]	Older adults aged 65 years and above.	The study team comprised two geriatricians and a geriatric nurse. They engaged primary care doctors to ensure follow-up care.	The study teams reported that the screening sessions took between 5 and 20 mins per person. The setting for the screening was a social club for older people run by the city council in Canillo (two geriatricians and one geriatrics nurse). A rolling recruitment process was used that followed a public health media campaign. The participants were screened and assessed as they were identified. The team was small and already had awareness of ICOPE, so it needed minimal training. The team assessed declines in IC at the community health facility. The team met with all participants to discuss the results of their screening and assessment. Any person with declines in mobility or cognition received an appointment with the lead geriatrician to develop a personalized care plan, which was then shared with the participant’s primary care doctor. More than half of the older persons had declines in cognition and more than one in three had depressive symptoms.
	**Implementation of Steps 1–5** **(2 study teams/sites)**	**Study team/study site 1**			
		**ICOPE-Care Programme in Toulouse City, Occitanie, France from October 2019 to September 2028.** **Setting: Community (primarily) and primary care** **Mobile health technology: ICOPE app, ICOPE MONITOR app, ICOPE BOT (conversational chat bot), FRAILTY-ICOPE database, telemedicine platform**			
		Guyonnet 2020 [40] (Methodology article)	Older persons aged more than 60 years (a group for those aged 40–60 years could be included)	Patients (with or without caregiver’s help), research or clinical nurse	Between two annual visits, IC domains will be monitored every 4 months by using the ICOPE MONITOR app. Six IC domains are monitored (with or without the help of a caregiver) in either the application developed in collaboration with WHO (ICOPE Monitor app) or a web platform, or through a phone call by a Gerontopole’s trained research nurse. When declines are detected in the ICOPE Step 1, a phone call is organized by the research nurse within one week to confirm this decline and to investigate the causes in collaboration with the medical research team. Participants will be trained to monitor their IC during the baseline visit by the Gerontopole’ research team. The remote monitoring of IC will last the whole length of the research study for up to ten years. Once an IC decline is confirmed, participants will have a thorough clinical assessment and blood sampling. The information will allow the examination of some markers of aging at the time declines are detected. There will be a personalized prevention care plan to maintain function according to the recommendations from the WHO ICOPE programme for usual care.
		Piau et al., 2020 [41] (Narrative description of INSPIRE)		Self-assessment.	There is self-screening every 4 months. There could be app-guided and chat box-guided questions. A team of nurses will receive the results remotely on a dashboard. When a decline is detected between two self-assessments, the patient will be offered a face-to-face evaluation. There could also be clinical confirmation tests.
		Takeda et al., 2020 [42] (Methodology article)		Self-assessment.	Self- assessment is every 4 months and monitored by the Toulouse regional team for ageing and prevention. Once IC decline is identified, the older adults will go through clinical assessment and blood sampling.
		WHO Clinical Consortium on Healthy Ageing 2019 [16]		Self-assessment.	The programme will involve biological sampling and self-monitoring using the ICOPE screening tool after training with a health care worker. The follow-up period will be 4 months for self-monitoring, and annual and biannual visits by health care workers.
		De Souto Barreto 2021 [43] (Methodology article)		Primary care providers (in particular, community nurses).	Primary care providers, in particular community nurses, will use the ICOPE app for tablets/smartphones to implement the ICOPE pathway for each older adult. ICOPE Step 1 (i.e., screening for low IC levels) will be performed in all individuals. The other steps of the model may be performed according to available resources in the local care services/facilities. A modified version of Step 1 (with discrete/continuous IC variables) in the ICOPE app will be used for the remote monitoring of IC levels over time. All the data will be automatically transferred to a secure database and will be used to investigate the evolution of IC domains over time.
		Lafont et al., 2021 [44] (Abstract only (only the abstract was English translated))		Nurses and pharmacists.	Digital tools were developed.
		Mathieu et al., 2021 [45] (Abstract only (only the abstract was English translated))		Nurses and pharmacists.	Nurses and pharmacists were invited to integrate Step 1 in their practices. They attended a free webinar training if they wanted to.
		Tavassoli et al., 2021 [46] (Methodology article)		Self-assessment, assistance from caregivers, healthcare professionals.	Step 1 will use the mobile app for remote monitoring on a large-scale. Seniors would be invited to use the Step 1 every 4–6 months. If the senior could not carry out the self-assessment, a professional would intervene every 6 months to perform Step 1. Steps 2 and 3 will involve assessment by the nurse and/or general practitioner if Step 1 was abnormal, or if an alert was generated during follow-up. If the screening of IC was abnormal, Steps 2 and 3 were carried out by the GP or nurse trained in geriatrics. Step 4 will involve Digital medicine used in this programme to simultaneously monitor large populations regardless of where they lived and ensure the implementation and follow up of the personalised care plan that was proposed in Step 3. Step 5 will be engagement of the community. This step is part of the INSPIRE ICOPE-CARE programme and plays an important role in bringing together different actors and organisations.
		WHO Clinical Consortium on Healthy Ageing 2020 [17]		Self-assessment by older adults and healthcare professionals (community nurses, general practitioners)	All domains of IC were systematically monitored through ICOPE tools, which has been adapted in digital form to make remote and large-scale monitoring possible. Two tools were developed, the ICOPE MONITOR and the ICOPE BOT, a conversational robot. Both are connected to the Gérontopôle frailty database. Step 1 was done every 4–6 months by healthcare professionals or older people themselves. If deterioration in one or more domains of IC is identified, an algorithm alerts health care professionals so they can quickly intervene. If abnormalities were determined in Step 1, nurses used a telemedicine platform to refer the older person to a primary care provider. There was teleconsultation by professionals with expertise in geriatric care. Measures included screening and full assessments, comprehensive battery of geriatric clinical assessments, body composition, maximal tests (VO² max, isokinetic strength) and biospecimens (fluids and cells/tissues).
		ICOPE implementation pilot programme: findings from the ‘ready’ phase. Geneva, 2022 [14]		1711 health and care workers (1053 nurses, 245 pharmacists, 104 doctors) and 20 post office workers) are trained for Step 1.	Older people having the opportunity to be involved in their own carewas a key message of the campaign in Occitanie to encourage participants to self-screen. This engagement and recruitment were achieved through a multimedia campaign using flyers, posters, a film promoting the ICOPE Monitor mobile app, webinars, conferences, and interviews. The assessments were organized and conducted by primary care workers (physicians, nurses, physiotherapist) with participants who had a positive screening result in IC at a health facility or the participant’s home, and used digital tools. Development of care plans and any necessary follow up were referred to the primary care workers. There was also community engagement. Ninety-four percent of participants had potential declines in at least one domain and there were also high numbers of participants that experienced declines in multiple domains. Approximately two-thirds of the older persons had loss of mobility and over 40% had issues with malnutrition.
		**Study team/study site 2**			
		**A healthy ageing programme in Chaoyang, Beijing, China (slogan: ICOPE will bring older people happiness)** **Setting: Community (Urban, suburban, and rural areas of the Chaoyang district)**			
		WHO Clinical Consortium on Healthy Ageing 2019 [16]	Community-dwelling persons aged 80 years and older with a risk of decline in IC, determined by ICOPE screening.	Not specified.	It involved ICOPE screening with examination of biomarkers. The next steps included validation of the screening tool including the app, a pilot study onintegrated interventions for specific IC abnormalities with different outcome measures, and the development of a project evaluation.
		WHO Clinical Consortium on Healthy Ageing 2020 [17]		Integrated care managers (may include community health workers. They are being trained to understand the care plan and deliver the ICOPE guidance.	The intervention will involve a personalized care plan with health and social care services coordinated by integrated care managers. The follow up period is 6–12 months. A longer study will be considered if findings demonstrate efficacy. Lower age groups may be enrolled.
		Zhao et al., 2021 [47]		Trained clinicians	IC domains included locomotion, vitality, sensory (vison and hearing impairments), cognition, and psychology. At the 1-year follow up, there was a stronger association between IC declines and dependency in ADL than the association between multimorbidity and ADL dependency.
		ICOPE implementation pilot programme: findings from the ‘ready’ phase. Geneva. 2022 [14]		Healthcare workers.	Step 1 was at the homes of older people and health centres in Chaoyang. Step 2 was carried out by integrated care managers with all participants. After developing personalized care plans with older people, integrated care managers (trained health and care workers) provided follow up sessions mainly through video calls. These aimed to support rehabilitative exercises, medication adherence and assistive care, and to check for any new or additional needs for social and health services. About 14% of respondents showed potential cognitive decline at screening compared with 37% at the formal assessment. A lower proportion of older adults had depressive symptoms.

AMICOPE: Aptitude Multi-domain group-based intervention to improve and/or maintain Intrinsic Capacity in Older People; CRELES: Longevity and Healthy Aging Study Costa Rica; IC: Intrinsic capacity; ICOOP_Frail: Integrated Care Of Older Patients with Frailty in Primary Care; MAPT: The Multidomain Alzheimer Preventive Trial.

## Data Availability

The corresponding author can be contacted on this matter.

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
