# Peer review of "The World Health Organization (WHO) Integrated Care for Older People (ICOPE) Framework: A Narrative Review on Its Adoption Worldwide and Lessons Learnt"

_ijerph, 2022, doi:10.3390/ijerph20010154_

Round 1

Reviewer 1 Report

Dear authors,

Thank you very much for your manuscript to review the ongoing studies on WHO Integrated Care for Older People globally. It is indeed very important to capture where we are, and learn from the lessons from different sites.

Major points:

1. ICOPE care pathways (handbook) describes the step wise approach, as author described, and WHO emphasizes the importance of whole steps, as Screening without assessment or intervention is harmful for older people. Please reflect this around lines 70-72 as well as discussion 2(line 184-199). Without consideration of the whole care pathway including monitoring and follow up, it will not capture 'feasibility' as a whole. 

2. ICOPE settings and objectives: There are discussion around the settings of ICOPE (line 168-183), but it is not clear to me what is the key message. ICOPE is for community and PHC, assuring the reference to secondary or specialized care, when needed. The objective of ICOPE is to prevent the declines in IC and maintain the IC. I would recommend the narrative of this section. 

3. Although inclusion criteria does not include the outcomes, it is helpful to include the brief outcome in the table 2, especially for feasibility phase and implementation phase. 

4. Lack of workforce (number, skills, incentive) is always highlighted from the ICOPE pilot sites. Please include the data/discussion on this subject. 

Minor points:

1. IC currently highlights 6 domains including cognitive capacity. Please include it in line 48-49.  

2. In the ICOPE approach, WHO does not call older people as 'patients', as declines in IC is not always considered as disease. Please replace 'patient' by 'person'

3. 'The actions required at system and service levels for high and low-and-middle income countries (LMICs) to adopt ICOPE were examined recently using a global Del- phi consensus study (line 64-64) ' This study has informed the WHO ICOPE implementation framework, published in 2019. 

4. Not many people understand the ICOPE step 1-5. Please include the figure to explain what are these steps (not just by text). 

Author Response

Please see attachment, thank you.

Reviewer 2 Report

Introduction: the description of the current status of the study is not very clear. Moreover, the differences between the content of the review of this study and the WHO report on the implementation of ICOPE are not clearly stated. The significance of this study is not very clear.

Please search more other countries' sub-centers of WHO websites.

The literature search did not use a combination of subject terms and free terms.

Why are the search start and end dates November 31, 2017? No reason was
given.

It is recommended that the study population should be defined as older adults (Table 1).

Data extraction and quality assessment: no description of data extraction.

Please add a description of the results to let the paper more meaningful and valuable.

The flowchart "Systematic evaluation and meta-analysis" is written in the title of Table 1 and please revise it.

The discussion section lacks an in-depth discussion of the results of the study. Please compare the results with existing results, analyze the reasons, and put forward constructive suggestions. 

Author Response

Please see attachment, thank you.

Reviewer 3 Report

This is a technically sound and useful narrative review providing an overview of 'early adopters' of the ICOPE framework. It is certainly not the fault of the authors if there are only very few of these, and that the implementation is rather piecemeal and often far from being 'integrated'. However, the authors have well synthesized potential learnings and implications for further research. As usual in this kind of exercises, restriction to the English language is a limitation, which has also been underlined by the authors. 

It is of course a bit unfortunate to use "IC" as an abbreviation for 'intrinsic capacity' in this article that is actually addressing 'integrated care' which has been abbreviated in other contexts with 'IC', too ... this is a bit confusing, but perhaps less for readers who are usually not dealing with integrated care.

There are some typos to be checked (e.g. lines 225 and 226). Is there something missing in line 230?

Finally, what I'm missing is a kind of critical assessment of the ICOPE framework itself - if the framework, in reality, is only adopted partly and generally very scarsely, couldn't this also be a sign that the framework itself has its caveats? And if yes, which? Some considerations regarding this aspect could contribute to making this article more interesting.

Author Response

Please see attachment, thank you.
